# StreamingKV: Adaptive Low-Rank KV Caching with Test-time Updates

## Abstract

Modern large language models (LLMs) face severe memory bottlenecks during inference due to the ever-growing key-value (KV) cache, especially in long-context settings. While recent low-rank compression techniques mitigate this issue, their reliance on static projection bases leads to suboptimal generalization across diverse prompts—ultimately compromising model performance. We introduce StreamingKV, an adaptive compression framework that dynamically updates the low-rank projection bases during inference, inspired by the Generalized Hebbian Algorithm (GHA). Unlike static methods, StreamingKV tailors the projection subspace to each input prompt in real-time, significantly enhancing representation quality with minimal computational overhead. Extensive experiments across multiple model families on long-context tasks demonstrate that StreamingKV consistently improves accuracy under the same compression ratio with negligible latency increase.

## 1 Introduction

Large Language Models (LLMs) have become a central focus in modern AI research due to their broad applicability across tasks. However, scaling LLMs to handle longer contexts and larger batch sizes introduces significant memory overhead, primarily from caching key–value (KV) representations during inference. To address this, architectural innovations such as state-space models (Gu & Dao, 2023) and linear RNN–based models (Peng et al., 2023), which offer linear time and memory complexity, have been proposed. On the attention front, methods like Grouped Query Attention (Ainslie et al., 2023) and Multi-Latent Attention (Liu et al., 2024a) reduce memory usage. Nevertheless, these approaches typically require modifying core architecture or attention layers and thus necessitate pretraining from scratch.

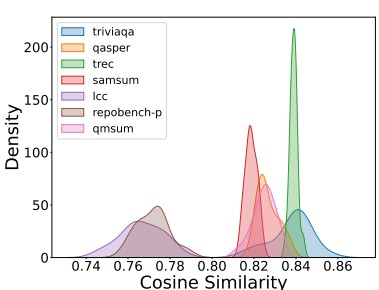

Figure 1: Reconstruction cosine similarity distribution per dataset of Llama-2-7B, 25[th] value projection layer. Cosine similarity varies significantly across datasets, indicating that a dataset-specific reconstruction strategy is necessary for preserving semantic fidelity.

To avoid pretraining, post-training or fine-tuning techniques have been widely explored. Strategies such as token-level selection/eviction (Zhang et al., 2023; Li et al., 2024; Xiao et al., 2024; Liu et al., 2023b), merging (Liu et al., 2024b), quantization (Hooper et al., 2024; Liu et al., 2024c), and low-rank compression (Chang et al., 2025; Sun et al., 2024; Zhang et al., 2024) aim to reduce KV cache size while preserving quality. Among them, low-rank token compression is attractive as it retains all tokens by projecting tokens into a compact subspace and reconstructing them when needed.

A practical challenge, however, is that most low-rank pipelines are formulated *offline*: they precompute a projection basis and keep it fixed throughout inference. In real deployments, inference is inherently *streaming*; topics, styles, and tasks drift within and across sessions. Therefore, a single static basis can mismatch evolving hidden-state distributions. Empirically, we observe reconstruction variability across datasets and layers under fixed-rank projections (e.g., dispersion in cosine similarity in Fig. 1); in addition, a Fisher-overlap analysis (Fig. 3) indicates that higher-frequency components are under-represented by a fixed basis, harming semantic fidelity when prompts deviate from pretraining.

Figure 2: Comparison to static low-rank compression and visualization of our method. (a) Static low-rank compression does not change compression layer weights; (b) StreamingKV dynamically adapts the weights as input prompt comes in, thereby becoming more prompt-aware.

Motivated by these observations, we propose **StreamingKV**, an adaptive compression framework that updates the projection bases at test time using a lightweight Generalized Hebbian Algorithm (GHA)–based rule (Sanger, 1988). StreamingKV dynamically adjusts the bases in response to the input prompt (Fig. 2): during the prefill stage, it aligns to the semantic structure of the incoming prompt; during decoding, it performs periodic, lightweight updates so that compressed representations remain aligned with the evolving token stream. The updates are decorrelating, compatible with standard low-rank decompositions, and introduce minimal computational overhead.

This adaptive mechanism improves fidelity at a fixed compression ratio while adding negligible latency and memory overhead, providing a practical path to long-context serving without architectural changes or retraining. We evaluate StreamingKV across multiple model families on long-context benchmarks and observe consistent preservation or improvement of task accuracy under static compression settings, alongside comparable memory usage.

Our contributions can be summarized as follows:

- We provide the first systematic analysis of why static-basis low-rank projections fail under distribution shift, using cosine-similarity dispersion and Fisher-overlap metrics to reveal the need for adaptation toward higher-frequency directions.
- We introduce **StreamingKV**, a test-time adaptive mechanism that leverages a lightweight GHA-based update to refine projection bases on the fly, while remaining fully compatible with standard low-rank decompositions.
- Experiments across long-context benchmarks show that StreamingKV consistently improves or preserves accuracy at the same compression ratio, with negligible computational overhead.

## 2 BACKGROUND AND PRELIMINARIES

### 2.1 KV CACHE AND MULTI-HEAD ATTENTION

The core operation of the Transformer architecture is the multi-head attention (MHA) mechanism (Vaswani et al., 2017). When a new input token at time $t$, $\mathbf{x} \in \mathbb{R}^d$, is introduced, it is linearly projected into query, key, and value representations using $h$ heads of learned weight matrices:

$$\mathbf{q}_i = \mathbf{x}\mathbf{W}_i^q, \quad \mathbf{k}_i = \mathbf{x}\mathbf{W}_i^k, \quad \mathbf{v}_i = \mathbf{x}\mathbf{W}_i^v \tag{1}$$

where $\mathbf{W}_i^q, \mathbf{W}_i^k, \mathbf{W}_i^v \in \mathbb{R}^{d \times d_h}$ are the projection matrices of $i$-th head, and $d_h$ is the dimensionality of each head. For multi-head attention with $h$ heads, $\mathbf{W}_q = [\mathbf{W}_1^q, \dots \mathbf{W}_h^q]$, $\mathbf{W}_k = [\mathbf{W}_1^k, \dots \mathbf{W}_h^k]$, and $\mathbf{W}_v = [\mathbf{W}_1^v, \dots \mathbf{W}_h^v]$. This means that the input is projected independently $h$ times, yielding sets $\{\mathbf{q}_i, \mathbf{k}_i, \mathbf{v}_i\}_{i=1}^h$. For each head $i$, the key and value states at time $t$ are obtained by concatenating all past tokens:

$$\mathbf{K}_i^{(t)} = [\mathbf{k}_{1,i}; \mathbf{k}_{2,i}; \dots; \mathbf{k}_{t,i}] \in \mathbb{R}^{t \times d_h}, \quad \mathbf{V}_i^{(t)} = [\mathbf{v}_{1,i}; \mathbf{v}_{2,i}; \dots; \mathbf{v}_{t,i}] \in \mathbb{R}^{t \times d_h}. \tag{2}$$

Then, each head computes scaled dot-product attention as follows:

$$\mathbf{a}_i = \text{Attention}(\mathbf{q}_i, \mathbf{K}_i, \mathbf{V}_i) = \text{Softmax}\left(\frac{\mathbf{q}_i \mathbf{K}_i^\top}{\sqrt{d_h}}\right) \mathbf{V}_i. \tag{3}$$

This formulation makes it clear that as sequence length $t$ grows, both $\mathbf{K}_i^{(t)}$ and $\mathbf{V}_i^{(t)}$ increase linearly in size, requiring $O(t \cdot d_h)$ memory per head. The concatenation of all past keys and values is therefore the source of the memory bottleneck in long-context inference.

The outputs of all heads are concatenated and passed through a final linear transformation:

$$\text{MHA}(x) = \text{Concat}(\mathbf{a}_1, \ldots, \mathbf{a}_h)\mathbf{W}_o, \tag{4}$$

where $\mathbf{W}_o = [\mathbf{W}_1^o, \ldots, \mathbf{W}_h^o] \in \mathbb{R}^{h \cdot d_h \times d}$ is the output projection matrix. This process allows the model to attend to information from multiple representation subspaces jointly.

## 2.2 LOW-RANK APPROXIMATION VIA SVD IN KV CACHE

Singular Value Decomposition (SVD) is a widely adopted method for obtaining low-rank approximations of matrices (Wall et al., 2003). Given a matrix $\mathbf{W} \in \mathbb{R}^{d \times d_h}$, SVD factorizes it into three components: $\mathbf{W} = \mathbf{U}\boldsymbol{\Sigma}\mathbf{V}^\top$, where $\mathbf{U} \in \mathbb{R}^{d \times d}$ and $\mathbf{V} \in \mathbb{R}^{d_h \times d_h}$ are orthogonal matrices containing the left and right singular vectors, and $\boldsymbol{\Sigma} \in \mathbb{R}^{d \times d_h}$ is a diagonal matrix of singular values.

To construct a rank-$r$ approximation, only the top $r$ singular values and corresponding singular vectors are retained:

$$\mathbf{W} \approx \mathbf{A}\mathbf{B}, \quad \mathbf{A} = \mathbf{U}_r \sqrt{\boldsymbol{\Sigma}_r}, \quad \mathbf{B} = \sqrt{\boldsymbol{\Sigma}_r}\mathbf{V}_r^\top \tag{5}$$

Here, $\boldsymbol{\Sigma}_r \in \mathbb{R}^{r \times r}$ contains the largest $r$ singular values, while $\mathbf{U}_r \in \mathbb{R}^{d \times r}$ and $\mathbf{V}_r \in \mathbb{R}^{d_h \times r}$ are the truncated singular vectors. This decomposition reduces the storage requirement from $d \cdot d_h$ to $r(d + d_h)$, making it especially effective for compressing large matrices.

Chang et al. (2025) utilizes this mechanism to make low-rank projection matrices for compressing keys and values as follows:

$$\mathbf{h} = \mathbf{A}^\top \mathbf{x}, \quad \mathbf{y} = \mathbf{B}\mathbf{h} \tag{6}$$

where $\mathbf{A} \in \mathbb{R}^{d \times r}$, $\mathbf{B} \in \mathbb{R}^{r \times d_h}$, $\mathbf{x}$ is input tokens and $\mathbf{y}$ is output representation reconstructed via SVD. Considering computational overhead, they demonstrated constructing SVD bases over groups of heads, called G-LRD, rather than across all heads at once, enables efficient low-rank projection while preserving model performance. In this setting, the weight matrices are formulated as:

$$\mathbf{W}_{g_j} \approx \mathbf{A}_{g_j} \mathbf{B}_{g_j} \tag{7}$$

where $j$ denotes group index, $\mathbf{W}_{g_j} = [\mathbf{W}_{j,1}, \ldots \mathbf{W}_{j,s}] \in \mathbb{R}^{d \times d_h \cdot s}$ denotes a group of $s$ heads, $\mathbf{A}_{g_j} \in \mathbb{R}^{d \times r_g}$ and $\mathbf{B}_{g_j} \in \mathbb{R}^{r_g \times d_h \cdot s}$. For reconstruction, the original key or value for each head can be reconstructed via:

$$[\mathbf{y}_{j,1}, \ldots \mathbf{y}_{j,s}] = \mathbf{h}_{g_j} \mathbf{B}_{g_j} \tag{8}$$

This group-wise setting balances between computation overhead and approximation accuracy.

## 2.3 GENERALIZED HEBBIAN ALGORITHM

The Generalized Hebbian Algorithm (GHA) (Sanger, 1988) is a biologically inspired online learning rule designed to extract the principal components of a data distribution in a streaming fashion. It extends Oja's rule (Oja & Karhunen, 1985), which itself is a normalization of Hebbian learning, to compute multiple orthogonal principal components through a simple neural-style iterative update.

Formally, given an input vector $\mathbf{x} \in \mathbb{R}^d$, GHA learns a set of orthonormal projection directions $\mathbf{W} \in \mathbb{R}^{r \times d}$ such that the projections $\mathbf{y} = \mathbf{W}\mathbf{x} \in \mathbb{R}^r$ approximate the top-$r$ principal components of the input distribution. The update rule for each timestep $t$ is given by:

$$\Delta \mathbf{W} = \eta \left(\mathbf{y}\mathbf{x}^\top - \text{tril}(\mathbf{y}\mathbf{y}^\top)\mathbf{W}\right) \tag{9}$$

where $\eta$ is the learning rate and $\text{tril}(\cdot)$ denotes the lower-triangular part of a matrix, ensuring decorrelation (and eventual orthogonality) among the learned components.

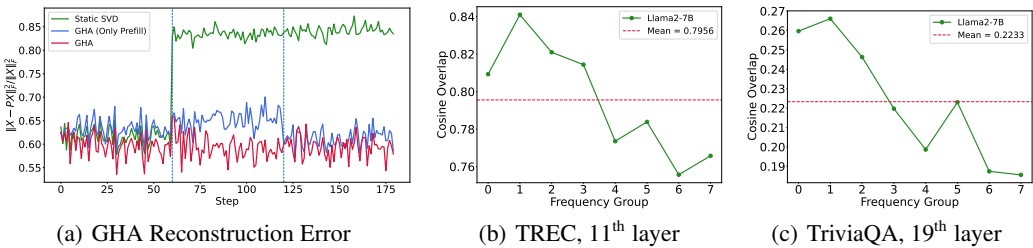

(a) GHA Reconstruction Error     (b) TREC, $11^{th}$ layer     (c) TriviaQA, $19^{th}$ layer

Figure 3: (a) Reconstruction error over segments on synthetic signal. The vertical lines indicate signal changes. (b) Fisher Overlap analysis of TREC on Llama-2-7B key projection layers. (c) Fisher Overlap analysis of TriviaQA on Llama-2-7B key projection layers.

## 3 MOTIVATIONAL ANALYSES

### 3.1 FISHER OVERLAP ANALYSIS ON SVD BASES

The key cache in transformer models is known to exhibit low-rank structure and significant redundancy (Kim et al., 2024; Zhang et al., 2024; Chang et al., 2025). Leveraging this observation, existing studies have explored low-rank compression techniques to improve memory efficiency. However, aggressively eliminating redundancy can lead to a degradation in performance due to excessive information loss. To better understand this trade-off, we employ Fisher overlap analysis to quantify the discrepancy between the pretraining dataset and downstream test datasets.

Following the methodology of Kirkpatrick et al. (2017), we investigate whether different tasks performed by the same model rely on overlapping subsets of parameters by analyzing the similarity between their respective Fisher information matrices. We first compute Fisher information matrix over the pretrained parameters of Llama-2, trained on CommonCrawl, Wikipedia and related datasets, using WikiText-2. Then we get Fisher information matrix with downstream test datasets in same mechanism. In this way, we respectively get $F_1$ and $F_2$ from WikiText-2 dataset and LongBench test datasets. Then, we decompose the pre-trained parameters of Llama-2 into their respective frequency bands and project computed Fisher information onto each frequency component band, represented as:

$$\tilde{F}_1^{(i)} = \mathbf{V}_i F_1 \mathbf{V}_i^\top \qquad \tilde{F}_2^{(i)} = \mathbf{V}_i F_2 \mathbf{V}_i^\top \qquad (10)$$

where $\mathbf{z}_i$ is $i$-th frequency band of a decomposed weight matrix. We use frequency band as bins of singular vectors by index. We then compute Fisher overlap with cosine similarity between the two datasets across these components.

$$\frac{\langle \tilde{F}_1, \tilde{F}_2 \rangle_F}{|\tilde{F}_1|_F |\tilde{F}_2|_F} \qquad (11)$$

As shown in Fig. 3, the overlap consistently decreases with higher frequency components, indicating that high-frequency components are more task-specific and less transferable from compressed weight matrix. Especially, in Fig. 3(c), TriviaQA datasets shows lower Fisher overlap scale, which means the dataset shows little alignment to pretrained training corpus. These observations suggest that updating the weight matrices along high-frequency components is essential for capturing task-specific information. For more visualizations, refer to Appendix H.

### 3.2 ADAPTIVE SUBSPACES: EFFICACY OF GHA

Unlike SVD, GHA requires no access to the full dataset or covariance matrix and the update rule ensures that the learned components remain approximately orthogonal over time. Under stationary input distribution and proper learning rate scheduling, the weight matrix $\mathbf{W}$ converges to the subspace spanned by the top-k eigenvectors of the input covariance matrix. These properties make GHA particularly suitable for adaptive subspace learning in non-stationary or resource-constrained settings. In the context of neural network compression and key-value caching, GHA can be used to refine low-rank projection bases in response to incoming data, without the need to recompute the full decomposition. We derive theoretical error bound of adding GHA to SVD in Appendix E.

An important question is whether the adaptive updates through GHA are essential and effective. To disentangle these effects, we compared three baselines in a controlled synthetic setting: (i) Static SVD, which fixes a global low-rank subspace estimated once and reuses it across all segments; (ii) GHA (prefill only), which updates the bases at each new segment; and (iii) GHA (prefill+decode), which performs lightweight updates throughout prefill and decode phase. The synthetic signal is piecewise-stationary: a latent k-dimensional subspace changes at the vertical lines in Fig. 3(a), while samples within a segment are i.i.d. from that subspace. For more details on experiment setting, refer to Appendix A.6. The results show that prefill-only GHA indeed reduces error compared to a global static basis, confirming the importance of prompt-specific initialization. However, without updates during decoding, the bases remain vulnerable to intra-segment drift, and reconstruction error accumulates. By contrast, reconstruction with decoding-time GHA updates consistently achieves the lowest error, maintaining stability even under distributional shifts. This highlights that the benefit of adaptive update lies not only in its ability to specialize per prompt but also in its capacity to adaptively refine token representations as inference progresses.

# 4 STREAMINGKV: TEST-TIME UPDATE USING GHA

## 4.1 INITIALIZATION

Since our approach is focused on enhancing performance of static low-rank projection KV caching method, we adopt the low-rank decomposition strategy proposed in (Chang et al., 2025). Following the approach used in SVD-LLM (Wang et al., 2025), we first construct the initial SVD bases from the model weights. We utilize a Hadamard matrix for efficient whiten-scaling.

$$\mathbf{W}_k \approx \mathbf{U}_k \mathbf{\Sigma}_k \mathbf{V}_k^\top = \mathbf{A}_k \mathbf{B}_k, \quad \mathbf{W}_v \approx \mathbf{U}_v \mathbf{\Sigma}_v \mathbf{V}_v^\top = \mathbf{A}_v \mathbf{B}_v \tag{12}$$

where $k$ and $v$ denotes matrix related to key and value state respectively. For efficient computation, we absorb reconstruction matrix for value $\mathbf{B}_v$ into the output projection matrix $\mathbf{W}_o$ offline, following previous works (Chang et al., 2025; Zhou et al., 2025; Zhang et al., 2024).

$$\mathbf{a}_i \mathbf{W}_i^o = (\mathbf{p}_i \mathbf{V}_i) \mathbf{W}_i^o = (\mathbf{p}_i \mathbf{h}_i^v \mathbf{B}_i^v) \mathbf{W}_i^o = \mathbf{p}_i \mathbf{h}_i^v (\mathbf{B}_i^v \mathbf{W}_i^o) \tag{13}$$

where $\mathbf{p}_i = \text{Softmax}\left(\frac{\mathbf{q}_i \mathbf{K}_i^\top}{\sqrt{d_h}}\right)$. Similarly, reconstruction matrix $\mathbf{B}_k$ for key is fused into the query projection matrix $\mathbf{W}_q$.

$$\mathbf{q}_i \mathbf{K}_i^\top = \mathbf{q}_i (\mathbf{h}_i^k \mathbf{B}_i^k)^\top = \mathbf{x}_t \mathbf{W}_i^q (\mathbf{B}_i^k)^\top (\mathbf{h}_i^k)^\top = \mathbf{x}_t (\mathbf{W}_i^q (\mathbf{B}_i^k)^\top)(\mathbf{h}_i^k)^\top \tag{14}$$

The optimal decomposition rank $r$ is selected based on Fisher information, calibrated using a small held-out dataset to guide rank selection following Chang et al. (2025). Specifically, each matrix's rank is determined by the ratio of its Fisher information to the total across all decomposed layers, allowing more informative components to preserve higher rank.

## 4.2 TEST-TIME UPDATE USING GHA

At inference time, once a prompt is received during inference, the basis is updated according to Algorithm 1 in Appendix D. Our primary focus is on improving the projection matrix $\mathbf{A}$, as it plays a key role in enhancing the expressiveness of the low-rank approximation. We first determine the target update rank $p$ based on a predefined high frequency ratio $\alpha$, i.e. $p = \lfloor \alpha \cdot r \rfloor$. Then, using GHA, we compute prompt-dependent bases updates and apply them to the existing projection weights. Here, we utilize lower-triangular matrix to ensure that the updates should be decorrelational to original basis, thereby obtaining only new information from newly updated components.

$$\Delta \mathbf{A} = \eta \cdot [\mathbf{x} \mathbf{h}^\top - \mathbf{A} \cdot \text{tril}(\mathbf{h} \mathbf{h}^\top)] \tag{15}$$

Given that attention outputs are the most sensitive in the high-frequency components of key and value caches from Section 3.1, our update strategy explicitly targets $p$ high-frequency components:

$$\hat{\mathbf{A}}_p = \mathbf{A}_p + \Delta \mathbf{A}_p, \quad \mathbf{h} = \hat{\mathbf{A}}^\top \mathbf{x} \tag{16}$$

where $\mathbf{A}_p \in \mathbb{R}^{d \times p}$ contains the singular vectors associated with the smallest $p$ singular values, representing the tail of the spectrum. We then use the updated weight matrix to project hidden states into a low-rank subspace, improving performance under the same memory budget.

Table 1: Experiment Results on LongBench. Experiments were conducted on 16 tasks; values are category-wise averages. The **Avg.** column is the overall average across all tasks.

| Model | Method | Comp. Ratio | Scores (↑) | | | | | | |
|---|---|---|---|---|---|---|---|---|---|
| | | | Summary. | Multi-QA | Single-QA | Few-Shot | Code | Synthetic | Avg. |
| Llama-2-7B | Full Cache | 1.00x | 14.09 | 7.11 | 15.58 | 65.17 | 63.25 | 4.04 | 27.53 |
| | Palu | 1.43x | 11.18 | **7.74** | 15.00 | 63.71 | 60.77 | 4.55 | 26.47 |
| | StreamingKV | 1.43x | **11.66** | 7.35 | **15.58** | **63.79** | **61.01** | **4.66** | **26.65** |
| | Palu | 2.00x | **9.60** | 8.00 | 15.81 | 60.39 | 48.71 | 4.64 | 24.39 |
| | StreamingKV | 2.00x | 9.34 | **8.51** | **16.42** | **61.52** | **48.97** | **5.21** | **24.73** |
| LongChat-7B-v1.5 | Full Cache | 1.00x | 26.75 | 24.05 | 31.35 | 64.77 | 54.98 | 15.25 | 36.14 |
| | Palu | 1.43x | 25.61 | 22.54 | 29.97 | 61.92 | 57.14 | 14.00 | 35.15 |
| | StreamingKV | 1.43x | **25.62** | **22.73** | **30.01** | **61.93** | **57.27** | 14.00 | **35.21** |
| | Palu | 2.00x | **23.14** | 22.73 | 25.73 | 59.74 | 40.62 | 6.46 | 30.51 |
| | StreamingKV | 2.00x | 23.09 | **22.80** | **25.75** | **59.99** | **40.77** | 6.46 | **30.58** |
| Mistral-7B-Instruct-v0.2 | Full Cache | 1.00x | 28.04 | 29.71 | 36.45 | 66.74 | 52.94 | 44.86 | 42.41 |
| | Palu | 1.43x | 27.40 | 30.18 | 36.61 | 66.04 | **52.28** | 37.92 | 41.33 |
| | StreamingKV | 1.43x | **27.46** | **30.56** | **36.98** | **66.19** | 52.05 | **37.98** | **41.48** |
| | Palu | 2.00x | **26.27** | 27.04 | 34.97 | 64.22 | 41.50 | **16.23** | 35.81 |
| | StreamingKV | 2.00x | 26.16 | **27.32** | **35.17** | **64.68** | **42.18** | 15.98 | **35.96** |

**Update granularity.** Although G-LRD introduces a group structure for computational efficiency at initialization, our update rule does not operate on groups independently. Whereas group-wise updates restrict adaptation to local partitions and may overlook cross-group correlations, joint-updates allows global coordination across all direction by updating the entire $r$-dimensional subspace jointly. In practice, the updates are implemented with parallel batched matrix multiplications on GPU, yielding the same asymptotic FLOPs while improving hardware utilization. From the perspective of subspace learning, this choice is theoretically sound: GHA is guaranteed to converge toward the top-$r$ eigenspace regardless of initialization, ensuring that adaptation follows the dominant directions of variation. Empirically, we find that such globally dependent updates improve downstream performance compared to independent group-wise update in Section 6.2.

## 5 EXPERIMENTS

### 5.1 SETUP

We assess StreamingKV across three large language model families: Llama-2 (Touvron et al., 2023), Mistral (Jiang et al., 2023), and LongChat (Li et al., 2023). For long-context evaluation, we use all available English tasks (sixteen tasks) from LongBench (Bai et al., 2023) and eleven tasks from RULER (Hsieh et al.). Unless otherwise noted, Full Cache refers to a model utilizing an uncompressed KV cache. Additional dataset and experimental setup details are provided in Appendix A.

### 5.2 EVALUATION ON LONG CONTEXT DATASETS

**LongBench** In Table 1, we conducted experiments on all available English tasks from the Long-Bench benchmark to check effectiveness of our method. Across the board, our method outperformed Palu on both 1.43x (30%) and 2.00x (50%) compression ratio on average for all settings, demonstrating improved performance under the same compression ratio. Notably, the model consistently demonstrates strong performance across the Single-QA, Multi-QA, and Few-shot categories. This observation may be attributed to the previously identified issue of low Fisher overlap between the test datasets and the pretrained weights and that our method successfully mitigated the issue. We first focus on Palu as the most closely related low-rank compression baseline, since it shares the same design philosophy of subspace projection as StreamingKV. Comparisons to other categories of approaches (e.g., eviction and quantization) are presented in Appendix B.1 for completeness.

**RULER** Table 2 presents results on the RULER benchmark with a 50% KV compression ratio. Across both Llama-2-7B and Mistral-7B-Instruct-v0.2, StreamingKV matches or surpasses Palu on nearly all tasks. Notably, under challenging settings such as MQ-NIAH and MV-NIAH, StreamingKV achieves consistent gains while maintaining parity on CWE and FWE. These results highlight the

Table 2: RULER benchmark result. On single NIAH (S-NIAH) and Multi-keys NIAH (MK-NIAH) task, we report average of three types of task. Avg. denotes average of all eleven tasks. Both Palu and StreamingKV results are demonstrated with 50% compression ratio.

| Model | Method | S-NIAH | MK-NIAH | MQ-NIAH | MV-NIAH | CWE | FWE | VT | Avg. |
|-------|--------|--------|---------|---------|---------|-----|-----|-----|------|
| Llama-2-7B | Full Cache | 94.3 | 62.67 | 86.45 | 79.65 | 71.38 | 82.13 | 47.36 | 76.18 |
| | Palu | 87.00 | 43.00 | 45.55 | 31.85 | **20.36** | **53.33** | 37.44 | 52.59 |
| | StreamingKV | **87.60** | **44.20** | **49.70** | **33.20** | 19.38 | 53.13 | **44.32** | **54.12** |
| Mistral-7B-Instruct-v0.2 | Full Cache | 98.80 | 81.07 | 95.80 | 94.15 | 45.18 | 88.07 | 94.00 | 86.98 |
| | Palu | **99.40** | 64.40 | 93.15 | 91.70 | **3.70** | 85.93 | **89.20** | 77.73 |
| | StreamingKV | **99.40** | **64.80** | **93.45** | **91.85** | 2.86 | **87.33** | 88.72 | **77.89** |

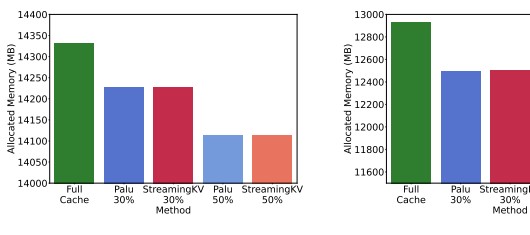

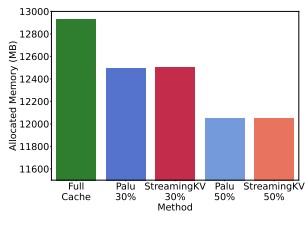

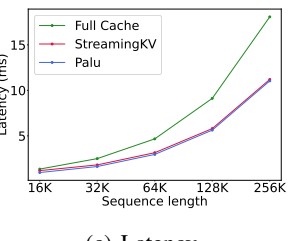

(a) Mistral-7B-Instruct-v0.2      (b) Llama-2-7B      (c) Latency

Figure 4: (a) Memory usage averaged over 10 test instances from the Qasper dataset, evaluated on the Mistral-7B-Instruct-v0.2 model after applying each compression method and (b) on Llama2-7B model. (c) Latency analysis of a single attention layer, averaged over 100 iterations.

robustness of prompt-adaptive updates: even when initialized from the same low-rank basis, GHA updates enable StreamingKV to recover better performance than the baseline.

## 5.3 MEMORY AND LATENCY ANALYSIS

We measured the memory usage required to process a single input sequence from the LongBench tasks. As shown in Fig. 4(a) and Fig. 4(b), our method maintains comparable memory consumption to the baseline, despite delivering improved performance. Detailed theoretical analysis of computation and memory complexity is provided in Appendix C.

In Fig. 4(c), we measured the latency of a single attention operation over varying sequence length from 16K to 256K. When sequence length is short, StreamingKV showed similar latency with full cache. However, as sequence length got longer, StreamingKV showed significant reduction compared to full cache, about 1.58x computational boost at sequence length 256K. Our method observed a minor overhead of approximately 0.3 ms from the basis adaptation step, an essential part of our method. However, the additional cost is negligible relative to the overall latency as sequence gets longer, and does not significantly impact the efficiency of the approach. This confirms that StreamingKV can achieve performance improvements with a simple method while incurring negligible time overhead. We defer end-to-end latency and throughput analysis to Appendix B.4.

## 6 DISCUSSION

### 6.1 FURTHER ANALYSES

In this subsection, we conducted a series of analyses ranging from reconstruction cosine similarity to alignment of weight matrices to evaluate the effectiveness of our method.

**Reconstruction cosine similarity** We measured cosine similarity between the original and reconstructed hidden states. In Fig. 5(a), StreamingKV achieves comparable cosine similarity to baseline on every layer. Specifically, in earlier layers of value projection, StreamingKV showed better cosine similarity to original hidden representation since value projection layer inherently shows more discrepancy. This stability across layers suggests that StreamingKV mitigates the degradation caused by distribution shift, ensuring more faithful information preservation throughout inference.

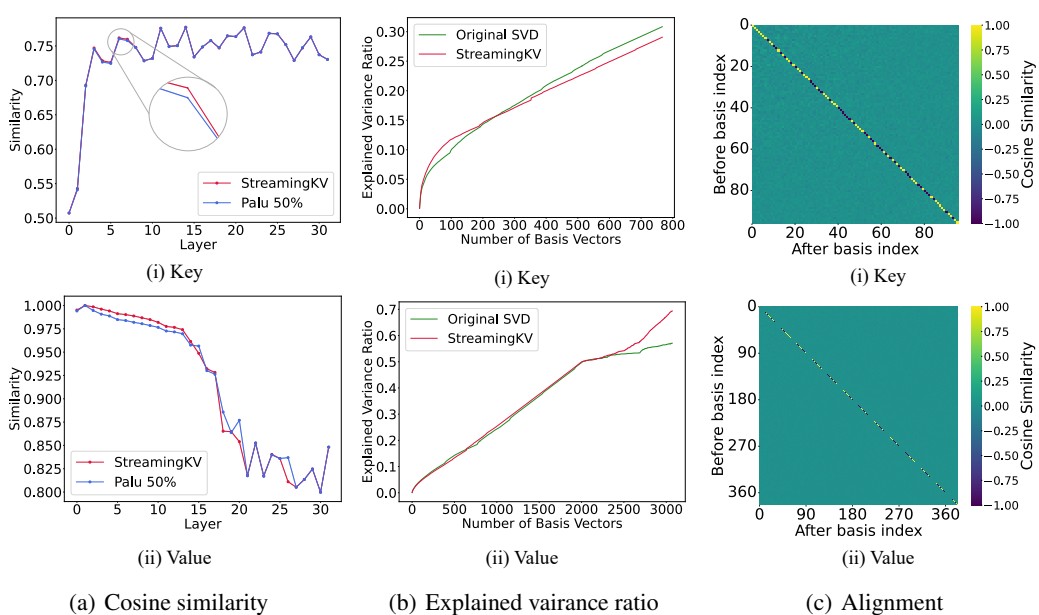

(a) Cosine similarity      (b) Explained vairance ratio      (c) Alignment

Figure 5: (a) Cosine similarity of reconstructed key and value states to original hidden states from baseline and StreamingKV. (b) Comparison of explained variance ratios for StreamingKV and original SVD and (c) Alignment between the updated weights from StreamingKV and the original weights obtained via SVD at the 16th layer of the Llama2-7B model.

Table 3: Ablation Studies. We test variants of our method on Mistral-7B-Instruct-v0.2 under a 50% compression ratio.

| Method | Multi-QA | Single-QA | Summary. | Few-Shot | Code | Synthetic | Avg. |
|--------|----------|-----------|----------|----------|------|-----------|------|
| w. tril | 27.19 | 35.02 | 26.03 | 64.10 | 41.88 | 16.23 | 35.83 |
| w.o tril | 26.89 | 34.83 | 26.15 | 64.22 | 41.41 | 16.48 | 35.75 |
| Joint | 27.25 | 35.07 | 29.39 | 64.28 | 26.12 | 15.98 | 35.90 |
| Group | 26.93 | 34.82 | 29.25 | 63.79 | 26.23 | 16.17 | 35.74 |

**Subspace representation quality**    We analyzed the subspace representation quality by examining the explained variance ratio of the projected hidden states from original weight and StreamingKV in Fig. 5(b). StreamingKV yields a higher explained variance ratio compared to the reconstruction from original SVD on Value side, indicating that a larger portion of the information is captured within the low-rank subspace. On key projection layers, StreamingKV gets higher explained variance ratio with less basis vectors at first, but original SVD gets more explained variance ratio as the number of basis vectors increases due to low-rank nature of the key. We attribute this improvement to the enhanced emphasis on high-frequency components during basis adaptation.

**Alignment of weight matrices**    In Fig. 5(c), we also evaluated the alignment between the original and reconstructed weights. The results show that the alignment is nearly perfect, even after the adaptation process. This indicates that our method preserves the structure of the original subspace while enabling more effective token representations by adaptively projecting each hidden representation to correct direction based on input token. Such preservation is crucial, as it suggests that the performance gains come not from altering the representational space destructively, but from adaptively refining it in a way that remains consistent with the model's original geometry.

## 6.2 ABLATION AND SENSITIVITY STUDIES

**Ablation studies**    We performed two ablation studies. First, we examined the effect of imposing a lower-triangular constraint on the update matrix. Compared to a full matrix, the constrained version yielded better performance, likely because its structured form enables more precise and efficient

Table 4: Sensitivity Studies. We report performance of different learning rates and high-frequency ratios tested on Mistral-7B-Instruct-v0.2 under a 50% compression ratio.

| Hyperparameter | Multi-QA | Single-QA | Summary. | Few-Shot | Code | Synthetic | Avg. |
|---|---|---|---|---|---|---|---|
| $\eta = $ 1e-5 | 27.20 | 35.02 | 29.59 | 64.49 | 26.17 | 15.98 | 35.94 |
| $\eta = $ 5e-6 | 27.25 | 35.07 | 29.39 | 64.28 | 26.12 | 15.98 | 35.90 |
| $\eta = $ 1e-6 | 27.23 | 35.13 | 29.31 | 64.37 | 26.16 | 15.98 | 35.92 |
| $\alpha = 0.2$ | 27.25 | 35.07 | 29.39 | 64.28 | 26.12 | 15.98 | 35.90 |
| $\alpha = 0.1$ | 27.14 | 35.15 | 29.41 | 64.29 | 26.05 | 15.98 | 35.88 |
| $\alpha = 0.05$ | 27.11 | 35.14 | 29.42 | 64.29 | 26.07 | 15.98 | 35.90 |

adaptation. Second, we compared joint versus group updates. Joint updates reduced reconstruction error and improved downstream performance (Table 3), suggesting that relaxing the group structure allows the model to capture cross-group correlations that static grouping would otherwise miss. Theoretically, GHA converges to the optimal top-$r$ eigenspace regardless of initialization, ensuring dependent updates remain aligned with subspace learning principles.

**Sensitivity studies**   To assess the sensitivity of our method to hyperparameters, we evaluated the effects of the learning rate $\eta$ and the high-frequency ratio $\alpha$ on Mistral-7B-Instruct-v0.2 with a 50% compression ratio. In Table 4, the results show that the method is relatively insensitive to changes in the learning rate, with overall performance remaining stable across different values. Similarly, variations in the high-frequency ratio $\alpha$ did not lead to significant performance differences. Interestingly, although the best performance on average was observed with tie at both $\alpha = 0.05$ and $\alpha = 0.2$, it varies by task category. This suggests that even if the updates are well-aligned with the prompt, the optimal proportion of high-frequency updates may vary by tasks. It also highlights that neither fully applying nor excessively suppressing high-frequency updates consistently leads to better performance, underscoring the need for a balanced update strategy.

## 7   RELATED WORK

**Low-rank compression in the KV cache**   Because keys (before RoPE) exhibit strong low-rank structure, a series of methods compress the KV cache via low-rank projections and reconstruct on demand. ShadowKV (Sun et al., 2024) compresses the key cache into a low-rank form and fetches value cache from CPU to extend context efficiently. Palu (Chang et al., 2025) leverages SVD to construct a projection layer that compresses hidden states and reconstructs them during inference. EliteKV (Zhou et al., 2025) introduces a RoPE-optimized scheme, selectively applying positional encoding only to high-frequency components. Unlike static SVD methods, our approach adaptively updates the projection subspace according to the input token stream, enabling more context-aware compression without compromising reconstruction quality. Similar line of work, Lexico (Kim et al., 2024), constructs sparse representations of keys and values using learned dictionaries.

**Online subspace learning**   Classical algorithms such as Oja's rule (Oja & Karhunen, 1985) have long been studied for their ability to update principal directions incrementally without full recomputation of covariance matrices. Recent theoretical advances refine these guarantees: for example, Jain et al. (2016) analyzes sample complexity and gap dependence for top-eigenvector estimation. Bhatia et al. (2018) extend Oja to generalized eigenvector settings, while Huang et al. (2021a) covers the multi-component case. These methods underpin the possibility of lightweight, online low-rank basis updates; similar in spirit to our use of the GHA for inference-time adaptation.

## 8   CONCLUSION

In this work, we introduced StreamingKV, an adaptive low-rank KV cache compression method that dynamically updates the projection bases during inference using GHA. By adaptively adjusting low-rank subspace, StreamingKV enables more accurate token representations while maintaining memory and computational efficiency. Our method outperforms prior approaches across a wide range of tasks and model families, demonstrating consistent improvements in long-context understanding with minimal additional overhead. Through empirical analysis, we show that adaptive updates better preserve semantic fidelity and critical high-frequency information in the key-value cache.

ETHICS STATEMENT

This work does not involve human subjects, private or sensitive data, or any proprietary datasets. All experiments are conducted on publicly available benchmarks such as LongBench. While improvements in long-context language modeling may enable more effective downstream applications, we acknowledge the general risks associated with large language models, including potential misuse for generating harmful or misleading content. We emphasize that our contributions are methodological and we encourage responsible use of this technology in alignment with the ICLR Code of Ethics.

LLM STATEMENT

We used a LLM to assist with polishing the writing of this paper. The LLM was employed for improving clarity, grammar, and style, but it did not contribute to the research design, methodology, or results. All ideas, analyses, and conclusions presented in the paper are solely those of the authors.

REPRODUCIBILITY STATEMENT

We provide the full implementation of our method at `https://anonymous.4open.science/r/StreamingKV-B3D5`. The link includes source code, evaluation scripts, and configuration files used in experiments. All datasets employed in this work are publicly available. We report experimental settings and detailed hyperparameters are listed in Appendix A. Experiments were conducted on NVIDIA A100 GPUs, and the provided README file contains step-by-step instructions to reproduce all results reported in the paper.

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

## A  EXPERIMENT DETAILS

### A.1  BASIC SETTINGS

Our experiments were conducted on a computing server equipped with an AMD EPYC 9654 CPU (2 sockets, 192 cores, 384 threads, 1.5–3.7 GHz, L3 cache 768 MiB) and a NVIDIA A100 80GB PCIe GPU with CUDA version 12.4.

### A.2  COMPRESSION SETTINGS

We built our implementation of StreamingKV using the Huggingface library (Wolf et al., 2020). For decomposing the key and value projection layers, we applied the truncation-aware SVD introduced in SVD-LLM (Wang et al., 2025). Our base low-rank compression method follows Palu's G-LRD approach (Chang et al., 2025). G-LRD was applied with a group size of 4, using the same rank size across all groups. To estimate Fisher information for rank selection, we sampled 1,024 random sequences from Wikitext-2, each with a length of 2,048 tokens.

### A.3  HYPERPARAMETERS

We report the best hyperparameter sets used for main table in Table 5. We search hyperparameter range for $\eta \in \{5e-5, 1e-5, 5e-6, 1e-6\}$, $\alpha \in \{0.05, 0.1, 0.2\}$, $n_{\text{iters}} \in \{2, 3\}$ and fixed $s = 10$.

Table 5: Best hyperparameter settings for StreamingKV across LongBench and RULER.

| Task | Model | $s$ | $n_{\text{iters}}$ | $\eta$ | $\alpha$ |
|---|---|---|---|---|---|
| LongBench | Llama-2-7B (30%) | 10 | 3 | 1e-6 | 0.2 |
| | Llama-2-7B (50%) | 10 | 3 | 1e-5 | 0.1 |
| | LongChat-7B-v1.5 (30%) | 10 | 3 | 1e-6 | 0.05 |
| | LongChat-7B-v1.5 (50%) | 10 | 3 | 1e-5 | 0.1 |
| | Mistral-7B-Instruct-v0.2 (30%) | 10 | 3 | 5e-6 | 0.1 |
| | Mistral-7B-Instruct-v0.2 (50%) | 10 | 2 | 5e-6 | 0.2 |
| RULER | Llama-2-7B (50%) | 10 | 3 | 1e-6 | 0.2 |
| | Mistral-7B-Instruct-v0.2 (50%) | 10 | 3 | 1e-5 | 0.1 |

### A.4  LONGBENCH EVALUATION DETAILS

For the LongBench (Bai et al., 2023) evaluation, we assessed all English-language tasks, covering sixteen benchmarks grouped into six categories to comprehensively evaluate StreamingKV. Task descriptions, abbreviation, and associated metrics are provided below. During inference, we set the maximum sequence length to 31,500 for both Mistral and LongChat, and 4,096 for Llama2.

- Single-Document QA
    - Qasper (Dasigi et al., 2021) (QA, F1 score)
    - NarrativeQA (Kočiskỳ et al., 2018) (NQA, F1 score)
    - MultiFieldQA-en (MQA, F1 score)
- Multi-Document QA
    - HotpotQA (Yang et al., 2018) (HQA, F1 score)
    - 2WikiMultihopQA (Ho et al., 2020) (2WM, F1 score)
    - MuSiQue (Trivedi et al., 2022) (MSQ, F1 score)
- Summarization
    - QMSum (Zhong et al., 2021) (QM, ROUGE score)
    - MultiNews (Fabbri et al., 2019) (MN, ROUGE score)
    - GovReport (Huang et al., 2021b) (GR, ROUGE score)
- Few-shot Learning

- – TREC (Li & Roth, 2002) (TR, classification score)
- – TriviaQA (Joshi et al., 2017) (TQA, F1 score)
- – SAMSum (Gliwa et al., 2019) (SS, ROUGE score)
- Code Completion
  - – LCC (Guo et al., 2023) (similarity score)
  - – RepoBench-p (Liu et al., 2023a) (RB, similarity score)
- Synthetic
  - – PassageCount (PC, Accuracy)
  - – PassageRetrieval (PR, Accuracy)

## A.5 RULER EVALUATION DETAILS

For the RULER (Hsieh et al.) benchmark, we evaluated across thirteen synthetic tasks spanning four categories: retrieval, multi-hop tracing, aggregation, and question answering. During inference, we set the maximum sequence length to 31,500 for Mistral and 4,096 for Llama2. We test for max context length 4K on Llama2, and 16K for Mistral. Below we list task categories, abbreviations, and associated evaluation metrics, along with brief descriptions.

- Retrieval
  - – Single NIAH (S-NIAH, Accuracy): Retrieve one "needle" (key-value pair) hidden in a long "haystack." Three needle types are used (Word–Number, Word–UUID, UUID–UUID), and the reported score is averaged over them.
  - – Multi-keys NIAH (MK-NIAH, Accuracy): Retrieve the correct value when multiple distractor keys are inserted. As with S-NIAH, three needle types are tested and averaged in reporting.
  - – Multi-values NIAH (MV-NIAH, Accuracy): Retrieve all values associated with a single key.
  - – Multi-queries NIAH (MQ-NIAH, Accuracy): Retrieve values for multiple distinct keys simultaneously.
- Multi-hop Tracing
  - – Variable Tracking (VT, Accuracy): Trace variable assignments through chains of references to find all variables pointing to the same value.
- Aggregation
  - – Common Words Extraction (CWE, Accuracy): Identify a fixed set of common words among distractors in long sequences.
  - – Frequent Words Extraction (FWE, Accuracy): Return the top-$K$ most frequent words sampled from a Zipfian distribution.

## A.6 SYNTHETIC SETTING FOR GHA ANALYSIS

To analyze the effectiveness of GHA in adaptive subspace tracking, we simulated a piecewise-stationary Gaussian stream. Each segment is generated from a $d = 40$ dimensional Gaussian distribution with a rank-$r = 6$ dominant subspace. Covariance matrices are constructed by random rotations with decaying eigenvalues (top eigenvalues ranging from 6.0 to 3.0, others fixed to 1.0). At each change point, the covariance is resampled with a different random seed, resulting in a new latent subspace. For each segment, we draw a prefill batch of 64 samples to initialize the subspace basis, followed by decode batches of 64 samples per step. Three baselines are compared:

- Static SVD: a single global low-rank basis estimated once from the first segment, reused thereafter.
- GHA (prefill only): re-estimates the basis with GHA at the start of each segment using prefill data, but remains fixed during decoding.
- GHA (prefill+decode): starts with prefill initialization and applies GHA update at each decoding step with learning rate $\eta = 2 \times 10^{-3}$. Bases are re-orthonormalized every 5 steps.

Table 6: Experiment Results on LongBench. Experiments were conducted on 16 tasks; values are category-wise averages. The **Avg.** column is the overall average across all tasks.

| Model | Method | Comp. Ratio | Scores (↑) | | | | | | |
|---|---|---|---|---|---|---|---|---|---|
| | | | Summary. | Multi-QA | Single-QA | Few-Shot | Code | Synthetic | Avg. |
| Llama-2-7B | Full Cache | 1.00x | 14.09 | 7.11 | 15.58 | 65.17 | 63.25 | 4.04 | 27.53 |
| | KIVI (2-bit) | 2.00x | 7.51 | 6.82 | 10.70 | 51.97 | 44.23 | 2.18 | 20.24 |
| | StreamingLLM | 2.00x | 7.47 | 6.52 | 12.37 | 61.59 | 64.35 | 3.81 | 25.01 |
| | SnapKV | 2.00x | 12.16 | 7.17 | 16.07 | 65.12 | 64.88 | 4.12 | 27.61 |
| | Palu | 2.00x | 9.60 | 8.00 | 15.81 | 60.39 | 48.71 | 4.64 | 24.39 |
| | StreamingKV | 2.00x | 9.34 | 8.51 | 16.42 | 61.52 | 48.97 | 5.21 | 24.73 |
| Mistral-7B-Instruct-v0.2 | Full Cache | 1.00x | 28.04 | 29.71 | 36.45 | 66.74 | 52.94 | 44.86 | 42.41 |
| | KIVI (2-bit) | 2.00x | 27.01 | 28.00 | 33.61 | 66.49 | 53.86 | 31.49 | 39.75 |
| | StreamingLLM | 2.00x | 27.74 | 29.74 | 36.01 | 66.72 | 55.87 | 44.66 | 42.61 |
| | SnapKV | 2.00x | 28.05 | 29.68 | 36.38 | 66.77 | 57.12 | 44.89 | 42.75 |
| | Palu | 2.00x | 26.27 | 27.04 | 34.97 | 64.22 | 41.50 | 16.23 | 35.81 |
| | StreamingKV | 2.00x | 26.16 | 27.32 | 35.17 | 64.68 | 42.18 | 15.98 | 35.96 |

Performance is measured by the normalized reconstruction error

$$E_t = \frac{\|X_t - \mathbf{W}_t \mathbf{W}_t^\top X_t\|_F^2}{\|X_t\|_F^2},$$

where $X_t$ is the current batch and $\mathbf{W}_t$ is the projection basis. Each segment consists of 60 decoding steps, and vertical dashed lines in Fig. 3(a) indicate distribution shifts.

## B    MORE EXPERIMENT RESULTS ON LONGBENCH

### B.1    COMPARISON WITH TOKEN EVICTION AND QUANTIZATION METHODS

We conducted additional experiments to compare StreamingKV against other categories of KV caching methods: token eviction and quantization. For a fair comparison, we configured all methods to operate under a comparable compression ratio. Specifically, token eviction approaches (StreamingLLM, SnapKV) were set to preserve half of the maximum context length (i.e., 50% of tokens), and the quantization baseline (KIVI) was applied with 2-bit precision.

Across most LongBench tasks, StreamingKV achieves performance comparable to token eviction–based approaches in Table 6. Interestingly, on Code and Synthetic tasks we observe a relative drop in performance. We attribute this to their reliance on exact token-level fidelity, where preserving tokens in their raw form (as in eviction) is advantageous. Our method, however, is motivated by the philosophy of reducing information loss through low-rank projection, which preserves all tokens in the cache rather than discarding them. In addition, we introduce adaptive updates oriented toward newly arrived tokens, allowing the model to maintain compact yet expressive representations over long contexts. This contrast highlights a fundamental trade-off: while our approach prioritizes holistic information preservation across all tokens, tasks that hinge on pinpointing a specific token may favor eviction-style strategies.

### B.2    SENSITIVITY ON HYPERPARAMETER SEARCH

We provide a more detailed evaluation of LongBench results here. Specifically, we report the scores across 16 tasks from LongBench. The mean and standard deviation are computed over different hyperparameter settings using Llama-2-7B with a 50% compression ratio. We aggregated twelve results over all hyperparameter search range.

### B.3    STATISTICAL SIGNIFICANCE ANALYSIS

To assess the robustness of our results, we conducted Wilcoxon signed-rank tests comparing StreamingKV against baseline across all LongBench tasks. We report two-sided $p$-values, rank-biserial correlation coefficients (effect sizes), and mean/median differences. We find that StreamingKV consistently improves over Palu with significant positive gain ($p$-value $< 0.05$). For Palu, the median gain of

Table 7: Mean and standard deviation of LongBench experiment over 12 tests.

| Tasks | QA | NQA | MQA | HQA | 2WM | MSQ | LCC | RB |
|-------|------|-------|-------|------|-------|------|-------|-------|
| Mean | 8.29 | 16.98 | 23.83 | 9.94 | 10.38 | 5.34 | 51.07 | 46.88 |
| Std. | 0.07 | 0.13 | 0.06 | 0.03 | 0.10 | 0.05 | 0.13 | 0.08 |
| **Tasks** | **QM** | **MN** | **GR** | **TR** | **TQA** | **SS** | **PC** | **PR** |
| Mean | 18.88 | 1.31 | 7.71 | 61.47 | 84.06 | 38.47 | 3.21 | 7.14 |
| Std. | 0.09 | 0.03 | 0.09 | 0.12 | 0.34 | 0.11 | 0.06 | 0.12 |

$+0.33$ points remains positive under a bootstrap-estimated 95% confidence interval $[+0.22, +0.42]$, further confirming the robustness of the observed advantage.

Table 8: Statistical significance of improvements over various baselines on LongBench. $\Delta$ denotes the score difference. $p$-values are from Wilcoxon signed-rank tests (two-sided). Effect sizes are reported as rank-biserial correlation coefficients ($r$).

| Model | # Tests | Mean $\Delta$ | Median $\Delta$ | Wilcoxon $W$ | $p$-val | Effect size $r$ |
|-------|---------|---------------|-----------------|--------------|---------|-----------------|
| Palu vs. StreamingKV | 192 | +0.29 | +0.33 | 5551 | 1.46e-6*** | +0.40 |

### B.4 ADDITIONAL LATENCY AND THROUGHPUT ANALYSIS

We report end-to-end generation latency and throughput on LongBench with Mistral-7B-Instruct-v0.2. We strictly maintain generated token length for all methods.

Table 9: End-to-end generation latency and throughput results on the LongBench task, measured and averaged over 10 samples. Throughput is measured in tokens per second.

| Method | Prefill Time (s) | Decode Time (s) | Total Time (s) | Prefill Thpt | Decode Thpt | Total Thpt |
|--------|------------------|-----------------|----------------|--------------|-------------|------------|
| Full KV | 1.366 | 1.166 | 2.539 | 9338.706 | 29.567 | 4686.501 |
| Palu (50%) | 1.369 | 1.198 | 2.574 | 9322.500 | 28.710 | 4617.536 |
| StreamingKV (50%) | 1.500 | 1.262 | 2.769 | 8438.028 | 26.932 | 4272.688 |

We report latency and throughput under identical prefill/decode settings, enforcing the same number of generated tokens across methods for a fair comparison. As shown in Table 8, StreamingKV exhibits slightly higher prefill and decode latency compared to Full KV and Palu. This overhead is expected since StreamingKV performs additional low-rank updates during decoding, which inevitably introduce extra computation. Nevertheless, the throughput remains within 80–90% of the baselines, indicating that the method is still practical for long-context inference. The small latency gap should be viewed in light of StreamingKV's ability to substantially reduce memory usage and maintain stable generation at such scales—capabilities that are not provided by the baselines.

## C COMPUTATIONAL AND MEMORY COMPLEXITY

We analyze the cost of one query step, where $t$ denotes the current context length, $h$ the number of heads, $d_h$ the per-head dimension, $g$ the number of groups, $r_g$ the low compression rank per group (so $r = g\, r_g$), and $s$ the GHA update interval (step size).

**Full KV caching** In the vanilla setting, computing attention scores requires the full $\mathbf{K}_i \in \mathbb{R}^{t \times d_h}$ and query $\mathbf{q}_i \in \mathbb{R}^{d_h}$, with cost

$$\mathcal{O}(t \cdot d_h \cdot h), \tag{17}$$

since the query of size $d_h$ attends over $t$ past keys per head. The memory requirement is

$$\mathcal{O}(t \cdot d_h \cdot h). \tag{18}$$

**Low-rank with reconstruction** With grouped low-rank decomposition of keys ($\mathbf{K} \approx \mathbf{B}h$ with per-group rank $r_g$), the per-step compute must include key reconstruction, RoPE re-application, and the subsequent GEMV. The resulting per-step cost is

$$\mathcal{O}(t \cdot r_g \cdot d_h \cdot h). \tag{19}$$

Only latent representations of rank $r_g$ per group are stored, so the memory is

$$\mathcal{O}(t \cdot r_g \cdot g). \tag{20}$$

**StreamingKV** We maintain a projector $W_g \in \mathbb{R}^{d_h \times r_g}$ per group and update it every $s$ steps using GHA update. For decoding ($t = 1$), one group update consists of

$$y = x^\top W_g \ (\mathcal{O}(d_h \cdot r_g)), \quad y^\top y \ (\mathcal{O}(r_g^2)), \quad x \cdot y^\top \ (\mathcal{O}(d_h \cdot r_g)), \quad W_g(y^\top y) \ (\mathcal{O}(d_h \cdot r_g^2)), \tag{21}$$

which is dominated by $\mathcal{O}(d_h \cdot r_g^2)$. Updating all $g$ groups once every $s$ steps yields the amortized per-step overhead:

$$\mathcal{O}\left(\frac{g}{s} \cdot d_h \cdot r_g^2\right). \tag{22}$$

Thus, the total per-step compute becomes

$$\mathcal{O}(t \cdot r_g \cdot d_h \cdot h) \ + \ \mathcal{O}\left(\frac{g}{s} \cdot d_h \cdot r_g^2\right). \tag{23}$$

**Relative overhead (vs. low-rank).** Compared to the low-rank per-step cost $\mathcal{O}(t \cdot r_g \cdot d_h \cdot h)$, the amortized GHA overhead ratio is

$$\frac{(g/s) \cdot d_h \cdot r_g^2}{t \cdot r_g \cdot d_h \cdot h} = \frac{g}{s \cdot h} \cdot \frac{r_g}{t} = \frac{r}{s \cdot h \cdot t}. \tag{24}$$

Hence, the incremental overhead vanishes as $t$ grows, and is further suppressed by a larger update interval $s$. In addition to the latent cache, GHA maintains projectors $W_g \in \mathbb{R}^{d_h \times r_g}$ for each group. These projectors are accessed only during update steps and not held permanently on GPU. These are constant-size and independent of $t$, so the asymptotic memory remains

$$\mathcal{O}(t \cdot r_g \cdot g). \tag{25}$$

Thus, our method preserves the memory reduction of low-rank caching, while adding only negligible constant overhead for the projectors.

**Implementation note.** For GPU efficiency we implement group updates via a single parallel matrix kernel (i.e., multiple groups fused into one batched GEMM). This preserves the same asymptotic FLOPs as the group-wise analysis above, but achieves higher hardware utilization in practice.

# D ALGORITHM FOR STREAMINGKV

In this section, we formularize StreamingKV as an algorithm in Alg. 1. StreamingKV applies test-time subspace updates both during the entire prefill phase and periodically during decoding. In decoding, the basis is updated once every 10 generation steps to balance adaptivity and efficiency. Unless otherwise noted, this schedule is fixed across all experiments. Updates are applied to both key and value projection matrices in every transformer layer, ensuring consistency of compression across the stack. We do not perform explicit re-orthonormalization after each update. In practice, our design re-initializes the token projection at every forward pass, which implicitly stabilizes the subspace geometry and prevents drift. While this deviates from the strictest form of online PCA, we found it maintains numerical stability in all tested long-context scenarios, and it avoids the additional computational overhead of periodic orthonormalization.

# E ERROR BOUND ANALYSIS

We analyze the error introduced by low-rank approximation in the KV cache. Our derivation follows the truncation-aware SVD analysis in Chang et al. (2025) and extends it to the adaptive basis used in StreamingKV.

---

**Algorithm 1** StreamingKV

---

**Input:** input tensor $\mathbf{x} \in \mathbb{R}^{B \times T \times d_h}$
**Parameters:** learning rate $\eta$, iterations $n_{\text{iters}}$, high-frequency ratio $\alpha$, projection $\mathbf{W} \in \mathbb{R}^{d_h \times r}$, step size $s$
**Output:** compressed tensor $\mathbf{x} \in \mathbb{R}^{B \times T \times r}$

1: **if** decode **and** $(\text{step} \bmod s \neq 0)$ **then**
2:      step $\leftarrow$ step $+1$
3:      **return**
4: **end if**
5: $p \leftarrow \lfloor r \cdot \alpha \rfloor$
6: $\mathbf{y} \leftarrow \mathbf{W}\mathbf{x}$
7: **for** $i \leftarrow 1$ **to** $n_{\text{iters}}$ **do**
8:      $\Delta\mathbf{W} \leftarrow \eta\big(\mathbf{y}^\top \mathbf{x} - \text{tril}(\mathbf{y}^\top \mathbf{y})\mathbf{W}\big)$
9:      Update last-$p$ rows of $\mathbf{W}$: $\mathbf{W}_{\text{tail}(p)} \leftarrow \mathbf{W}_{\text{tail}(p)} + \Delta\mathbf{W}_{\text{tail}(p)}$
10: **end for**
11: step $\leftarrow$ step $+1$ $=0$

---

**Setup.** Let $\mathbf{X} \in \mathbb{R}^{d \times n}$ denote the hidden states in a KV cache segment, with covariance $\mathbf{C} = \frac{1}{n}\mathbf{X}\mathbf{X}^\top$. Consider a linear map $\mathbf{W} \in \mathbb{R}^{n \times m}$ acting on these states and introduce a whitening transform $\mathbf{S}$ defined by the Cholesky decomposition of $\mathbf{X}^\top\mathbf{X} + \lambda\mathbf{I}$, so that $\mathbf{S}\mathbf{S}^\top = \mathbf{X}^\top\mathbf{X} + \lambda\mathbf{I}$. We can then rewrite

$$\mathbf{X}\mathbf{W} = (\mathbf{X}\mathbf{S}^{-1})(\mathbf{S}\mathbf{W}) = \tilde{\mathbf{X}}\,\tilde{\mathbf{W}},$$

with $\tilde{\mathbf{X}} = \mathbf{X}\mathbf{S}^{-1}$ and $\tilde{\mathbf{W}} = \mathbf{S}\mathbf{W}$.

**Truncation under SVD.** Let $\tilde{\mathbf{W}} = \mathbf{U}\boldsymbol{\Sigma}\mathbf{V}^\top$ be the singular value decomposition (SVD), and denote by $\tilde{\mathbf{W}}' = \mathbf{U}_r\boldsymbol{\Sigma}_r\mathbf{V}_r^\top$ its rank-$r$ truncation. The compressed weight is $\mathbf{W}' = \mathbf{S}^{-1}\mathbf{U}_r\boldsymbol{\Sigma}_r\mathbf{V}_r^\top$. For any input $\mathbf{X}$, the outputs before and after compression satisfy

$$\mathbf{Y} - \mathbf{Y}' = \tilde{\mathbf{X}}(\tilde{\mathbf{W}} - \tilde{\mathbf{W}}') = \tilde{\mathbf{X}}\,\mathbf{U}_T\boldsymbol{\Sigma}_T\mathbf{V}_T^\top,$$

where $T = \{r+1, \ldots, t\}$ indexes the truncated singular values. Assuming $\tilde{\mathbf{X}}^\top\tilde{\mathbf{X}} \approx \mathbf{I}$, the reconstruction error becomes

$$\|\mathbf{Y} - \mathbf{Y}'\|_F^2 \approx \sum_{i=r+1}^{t} \sigma_i^2, \tag{26}$$

which recovers the classical Eckart–Young theorem.

**Explicit upper bound.** Since $\|\boldsymbol{\Sigma}_T\|_F \leq \sqrt{s}\,\sigma_{\max}(\tilde{\mathbf{W}})$ with $s = t - r$, and $\sigma_{\max}(\tilde{\mathbf{W}}) \leq \|\mathbf{S}\|_2\|\mathbf{W}\|_2 = \sqrt{\sigma_{\max}(\mathbf{X}^\top\mathbf{X}) + \lambda}\,\|\mathbf{W}\|_2$, we obtain

$$\|\mathbf{Y} - \mathbf{Y}'\|_F \leq \sqrt{s}\,\sqrt{\sigma_{\max}(\mathbf{X}^\top\mathbf{X}) + \lambda}\,\|\mathbf{W}\|_2. \tag{27}$$

Thus, truncating $s$ singular values increases the error at most as $\mathcal{O}(\sqrt{s})$, consistent with empirical trends.

**StreamingKV via GHA.** Specializing to the hidden states themselves, let $\mathbf{U}_r \in \mathbb{R}^{d \times r}$ be the top-$r$ eigenvectors of $\mathbf{C}$. Then the optimal rank-$r$ reconstruction is Then the optimal rank-$r$ reconstruction is $\mathbf{X}_r = \mathbf{U}_r\mathbf{U}_r^\top\mathbf{X}$ with error $\|\mathbf{X} - \mathbf{X}_r\|_F^2 = \sum_{i=r+1}^{d} \sigma_i^2$.

StreamingKV replaces the static basis $\mathbf{U}_r$ with an adaptive basis $\hat{\mathbf{U}}_r(t)$ obtained from GHA updates. Define $\hat{\mathbf{X}}_r(t) = \hat{\mathbf{U}}_r(t)\hat{\mathbf{U}}_r(t)^\top\mathbf{X}$. The reconstruction error decomposes as

$$\|\mathbf{X} - \hat{\mathbf{X}}_r(t)\|_F^2 = \|\mathbf{X} - \mathbf{X}_r\|_F^2 + \epsilon_t, \tag{28}$$

where $\epsilon_t$ captures deviation between $\hat{\mathbf{U}}_r(t)$ and $\mathbf{U}_r$. Formally, letting $\mathbf{P} = \mathbf{U}_r\mathbf{U}_r^\top$ and $\hat{\mathbf{P}}(t) = \hat{\mathbf{U}}_r(t)\hat{\mathbf{U}}_r(t)^\top$,

$$\|(\mathbf{I} - \hat{\mathbf{P}})\mathbf{X}\|_F \leq \|(\mathbf{I} - \mathbf{P})\mathbf{X}\|_F + \|\hat{\mathbf{P}} - \mathbf{P}\|_2\,\|\mathbf{X}\|_F,$$

so that $\epsilon_t$ can be bounded in terms of $\|\hat{\mathbf{P}} - \mathbf{P}\|_2$. By Davis–Kahan theory, $\|\hat{\mathbf{P}} - \mathbf{P}\|_2 \leq \|\hat{\mathbf{C}} - \mathbf{C}\|_2/\delta$, where $\delta$ is the eigengap.

**Bound properties.** Classical results for GHA (Sanger, 1988) ensure that, with sufficiently small learning rate and ergodic sampling, $\hat{\mathbf{U}}_r(t)$ converges almost surely to the span of $\mathbf{U}_r$. Hence

$$\lim_{t \to \infty} \epsilon_t = 0, \tag{29}$$

and the worst-case bound of StreamingKV matches that of SVD:

$$\|\mathbf{X} - \hat{\mathbf{X}}_r(t)\|_F^2 \leq \sum_{i=r+1}^{d} \sigma_i^2 + \epsilon_t. \tag{30}$$

**Practical implication.** Static methods fix $\mathbf{U}_r$ once, which may mismatch new prompt distributions. In contrast, StreamingKV adapts $\hat{\mathbf{U}}_r(t)$ continuously, keeping $\epsilon_t$ small even under distribution shift. Thus, while the theoretical worst-case error is identical to SVD, the adaptive basis yields lower average reconstruction error across diverse prompts, explaining the empirical robustness of StreamingKV in long-context inference.

# F    EXTENDED RELATED WORKS

**Token eviction and quantization in the KV cache**    Token eviction strategies retain salient key–value pairs while pruning redundant entries. $H_2O$ (Zhang et al., 2023) introduces a memory-efficient eviction policy that selectively keeps high-impact tokens, improving throughput and latency. SnapKV (Li et al., 2024) automatically selects head-specific important KV positions based on attention patterns in a fixed observation window. Quantization methods, in turn, reduce bit-widths to minimize memory footprint and accelerate computation. KIVI (Liu et al., 2024c) analyzes the distinct characteristics of key and value representations during quantization and applies specialized strategies (channel-wise for keys, token-wise for values) to maximize efficiency while minimizing performance degradation. In short, prior art either (a) keeps a fixed low-rank basis (robust but sensitive to prompt drift), or (b) adapts token retention/quantization (adaptive but potentially lossy). StreamingKV explores the missing axis: online updates of the low-rank basis at test time, providing prompt-aware, lossless-oriented compression that complements existing approaches.

**Adaptive KV caching**    Recent efforts have explored adaptive strategies for KV cache management. LESS (Dong et al.) proposes a hybrid approach that combines sparse KV policies with low-rank states to preserve long-range dependencies while reducing memory usage. Ada-KV (Feng et al., 2024) addresses the limitations of uniform eviction by reallocating cache budgets adaptively across attention heads based on their distribution. Quest (Tang et al., 2024) introduces query-aware sparsity, dynamically selecting salient tokens in the KV cache to reduce latency while maintaining long-context accuracy. In contrast to these eviction- and sparsity-based approaches, our work focuses on adaptive low-rank compression, enabling the KV cache to adjust online to prompt variability without discarding tokens.

**Test-time adaptation**    Another closely related line is test-time adaptation (TTA), where models adapt to distribution shifts at inference without access to labeled data. For example, Tent (Wang et al., 2020) proposes fully test-time adaptation via entropy minimization, adjusting batch-norm statistics and affine parameters to better fit corruptions and domain shifts. Other works like EATA (Niu et al., 2022) introduce mechanisms to avoid degradation on original (in-distribution) data when adapting to out-of-distribution inputs. More recently, continual or lifelong TTA settings (Brahma & Rai, 2023) explicitly address streaming inputs over time, handling changing target domains and uncertainty, while incorporating regularization to preserve knowledge of earlier distributions. Moreover, Park et al. (2023) progressively adapt under target distribution shifts but risk forgetting earlier domain or source distribution performance; Fisher information has been used to select parameters or modulate updates to avoid catastrophic forgetting.

# G    LIMITATION AND FUTURE WORK

While StreamingKV achieves strong results on long-context benchmarks, several limitations remain. Our approach still requires tuning of hyperparameters such as learning rate and update ratio to get

optimal results. In addition, although the overhead is modest at the layer level, system-level latency under realistic serving scenarios has not been fully addressed. Future work will focus on developing automatic mechanisms for hyperparameter adaptation and optimizing the method for system-level efficiency, making it more suitable for deployment in practical serving environments. We also aim to validate StreamingKV across a wider variety of tasks beyond current benchmarks, which we believe will further demonstrate its robustness and applicability.

## H MORE VISUALIZATIONS

In this section, we provide more visualizations of analyses we conducted in main paper.

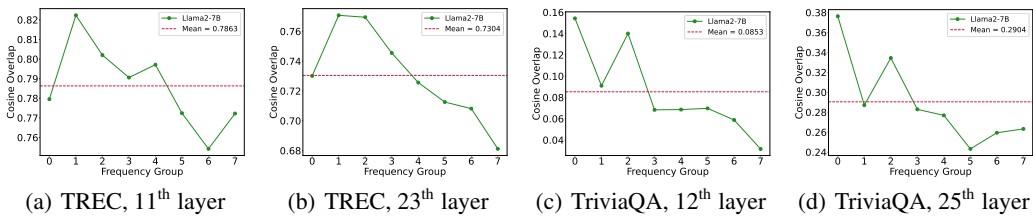

(a) TREC, 11th layer    (b) TREC, 23th layer    (c) TriviaQA, 12th layer    (d) TriviaQA, 25th layer

Figure 6: More visualization of Fisher Overlap analysis on TREC and TriviaQA.

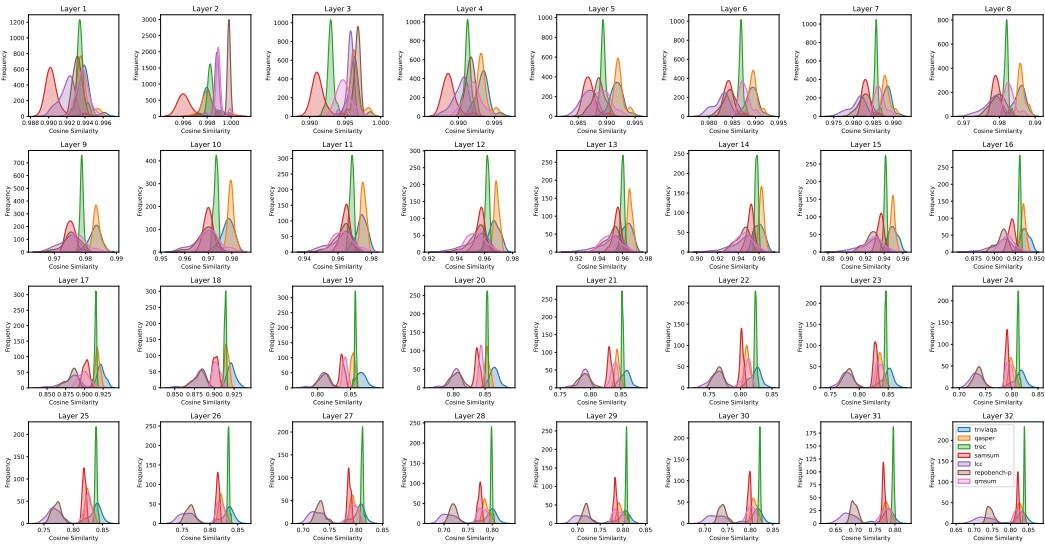

Figure 7: Reconsturction cosine similarity distribution per dataset of Llama-2-7B, compression ratio 50%, all value projection layers.

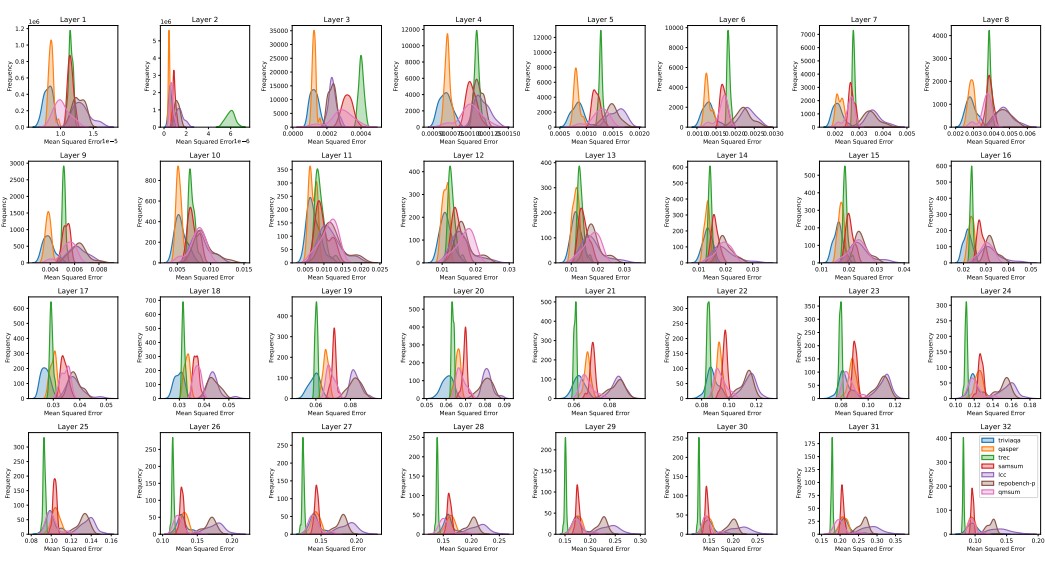

Figure 8: Reconsturction mean squared error distribution per dataset of Llama-2-7B, compression ratio 50%, all value projection layers.

