# OpenReview forum: "StreamingKV: Adaptive Low-Rank KV Caching with Test-time Updates"
_ICLR.cc/2026/Conference — ICLR 2026 Conference Withdrawn Submission_

### Official Review · Reviewer_z88f · 2025-10-30

**Soundness:** 1
**Presentation:** 3
**Contribution:** 2
**Rating:** 2
**Confidence:** 4

**Summary:**

This paper introduces an adaptive low-rank compression method for the LLM KV cache. The core innovation uses the Generalized Hebbian Algorithm to dynamically update static low-rank projection bases during both prefill and decode stages. While the method is novel and convincingly outperforms the static low-rank baseline, the overall contribution is significantly weakened by insufficient comparison against other methods. Additionally, important results are "hidden" in the appendix.

**Strengths:**

The method is well presented and justified:
-  The use of an online, lightweight algorithm (GHA) for dynamic basis adjustment at test time is novel and effectively addresses distribution shift during streaming inference.
- Strong Empirical Justification: The Fisher Overlap analysis provides evidence, clearly justifying why static bases fail and validating the necessity of continuous, decoding-time updates.

**Weaknesses:**

- Insufficient baselines: The comparison set is quite narrow. Only one baseline is showed in the main paper, and the the other baselines shown in the appendix are not very strong (streaming llm is one of the first KV compression methods, and SnapKV is also not recent).
- The additional baselines are relegated to the appendix, preventing a full and transparent assessment in the main paper.
- Fixed Compression Ratios: The performance is only evaluated at fixed compression ratios. To characterize the method's robustness and trade-off curve, the authors should evaluate performance across a diverse range of compression ratios.
- The crucial end-to-end throughput analysis is hidden in the Appendix and should be transparently discussed in the main body.
- The dependency on tuning both the learning rate  and high-frequency ratio for each model and compression ratio weakens the claim of an easily deployable, training-free method.

**Questions:**

- Integration of Baselines: Table 6 which includes methods from Quantization (KIVI) and Token Eviction (SnapKV, StreamingLLM), shows these methods are highly competitive, and in some cases, superior to StreamingKV. To validate the effectiveness of the proposed method, additional baselines such as DuoAttention or L2 Norm or PyramidKV should be included.

- Given that StreamingKV is an adaptive SVD-based method, what is the relation to other KV Cache compression methods and sparse attention methods ? How does this connect to e.g. QFilters or other SVD-based approaches ?

- The current evaluation is restricted to only two fixed compression ratios. The paper would benefit from performance results across a diverse range of compression ratios.

---

### Official Review · Reviewer_LReL · 2025-10-31

**Soundness:** 2
**Presentation:** 3
**Contribution:** 3
**Rating:** 4
**Confidence:** 3

**Summary:**

This paper introduces StreamingKV, a test-time adaptive framework for low-rank KV cache compression in large language models. Building on the Generalized Hebbian Algorithm (GHA), the method continuously updates the high-frequency components of SVD-based key/value projections during inference, enabling the cache to adapt to distributional shifts across prompts and tasks with negligible latency. The authors also propose a novel diagnostic metric, Fisher overlap, to quantify alignment between pretraining and downstream Fisher information across SVD frequency bands, revealing that high-frequency directions are more task-specific. Experiments on LLaMA-2-7B, Mistral-7B, and LongChat show consistent improvements over static SVD methods under 30–50% compression. The approach is simple, compatible with existing pipelines, and provides both empirical and analytical insights into adaptive KV compression.

**Strengths:**

1.  The paper identifies a concrete limitation in static low-rank KV compression—its inability to generalize across prompts due to fixed projection bases. The proposed StreamingKV introduces a test-time adaptive low-rank update mechanism based on the Generalized Hebbian Algorithm (GHA). This is a novel application of online subspace learning in the context of LLM inference, marking a conceptual advance beyond static SVD or Palu-style compression.
2. The paper introduces Fisher overlap, a new diagnostic metric that projects Fisher information matrices of the pretraining and downstream data distributions onto SVD frequency subspaces and measures their Frobenius-norm cosine similarity. This provides an interpretable, quantitative view of how “general” (low-frequency) and “task-specific” (high-frequency) components differ across distributions.

**Weaknesses:**

1. The paper mainly contrasts with SVD/Palu compression. It would strengthen the contribution to include more adaptive KV approaches like CommVQ or RAZORATTENTION.
2. While the paper reports consistent gains over static low-rank baselines, it remains unclear whether the improvement truly arises from the proposed high-frequency adaptive updates or simply from increased model flexibility. Two specific ablations would substantially strengthen the empirical evidence:
(1) Equal-Cost Rank Expansion Baseline: compare with a static SVD model of slightly larger rank but similar computational cost, to ensure the gain is not merely due to increased capacity.
(2) Random-vs-Structured Update: apply GHA updates to randomly selected directions instead of the designated high-frequency components, to verify that the structured frequency-based adaptation is essential.

**Questions:**

See Weaknesses. If the authors can adequately address the weaknesses noted above, I would be willing to raise my score.

---

### Official Review · Reviewer_uLpY · 2025-11-01

**Soundness:** 3
**Presentation:** 3
**Contribution:** 3
**Rating:** 4
**Confidence:** 3

**Summary:**

The paper introduces StreamingKV, an adaptive KV cache compression method. It uses the Generalized Hebbian Algorithm (GHA) to dynamically update low-rank projection bases during inference, aiming to improve performance on diverse prompts where static bases fail.

**Strengths:**

- the paper identifies a valid limitation of static low-rank compression methods, which is their inability to adapt to prompt-specific data distributions.

- the connection to test-time adaptation is novel and reasonable

- the motivational analyses, including Fisher overlap and synthetic data experiments, are interesting and informative

**Weaknesses:**

- The primary weakness is that the empirical performance improvements over the static baseline (Palu) are minimal across all reported benchmarks. The gains are often tenths of a percentage point, which is practically insignificant.

- The paper mentioned Oja's rule multiple times, it seems using Oja's rule would be an important deadline.

**Questions:**

The paper's motivation is to handle complex, dynamic contexts. Why not evaluate OjaKV on newer models with dedicated reasoning capabilities (e.g., Qwen3-Thinking) or on more recent, challenging reasoning benchmarks like AIME, which would truly test the method's ability to adapt?

---

### Official Review · Reviewer_eCnd · 2025-11-02

**Soundness:** 3
**Presentation:** 3
**Contribution:** 2
**Rating:** 2
**Confidence:** 5

**Summary:**

This paper proposes updating the low‑rank KV‑cache projection bases online (via a GHA/Oja‑style rule) to tailor compression to each input prompt at inference-time. Their method, StreamingKV, updates the projection bases at test time using a lightweight Generalized Hebbian Algorithm (GHA)–based rule, and these lightweight updates remain low-rank and adapt to the evolving token stream. They motivate their method by first providing a systematic analysis of how reconstruction loss increases for static-basis low rank projection methods. The idea is simple and seems straightforward to implement. Below is a brief algorithmic summary.

Given an input token embedding $x \in \mathbb{R}^{d}$, we first compute its low-rank latent representation $h = A^\top x \in \mathbb{R}^r.$
Then, StreamingKV performs a GHA (Oja–Sanger) style update on the last $p (<r)$ columns of $A$:
$$
\Delta A = \eta \left[ x h^\top - A \mathrm{tril}(h h^\top) \,\right],
$$
$$
A_p \leftarrow A_p + (\Delta A)_p \; h \leftarrow A^\top x
$$
where $\eta$ is the learning rate, $\mathrm{tril}(\cdot)$ denotes the lower-triangular operator, and $A_p$ refers to the last $p$ columns of $A$, corresponding to the least-significant singular components. During decoding, this update is applied once every $s$ tokens, while during prefill it is applied at every token.

Overall, the contributions can be summarized as:
- Diagnosis of static low‑rank failures under distribution shift using “Fisher overlap” computed per singular‑vector frequency band, showing cosine overlap drops for higher‑frequency components. showing the need for a method that adapts per-task to accomodate these high frequency components.
- Novel method StreamingKV that keeps the usual SVD initialization but adapts the projection basis at test time with a GHA update that targets the tail of the spectrum (small singular values), applied during prefill and periodically during decode.
- On LongBench and RULER, StreamingKV matches or slightly beats Palu at 30–50% compression across Llama‑2‑7B, LongChat‑7B‑v1.5, and Mistral‑7B‑Instruct‑v0.2 other low-rank projection methods such as Palu.

**Strengths:**

- Originality: The paper introduces a conceptually simple and relatively novel approach—test-time adaptive low-rank projection updates for KV-cache compression—by integrating a GHA-style update rule into inference. (I qualify the novelty as the paper builds largely on top of Palu)
- Quality: The proposed algorithm is lightweight and compatible with existing grouped-low-rank decompositions. The derivations and computational complexity analysis are mostly sound, and the experimental results spans multiple model families and benchmarks. Their Fisher-overlap analysis provides a reasonable motivation for StreamingKV.
- Clarity: The paper is quite readable and not too difficult to parse. The overall flow of the paper makes sense, though some important results and discussions were pushed to the appendix.
- Significance: KV cache compression is extremely important, and the paper proposes a deployable method with some meaningful results. It is important that the method does not occur much overhead, so while their results are not state-of-the-art, if the community adopts and builds on top of StreamingKV, stronger low-rank methods may emerge.

**Weaknesses:**

Major weakness:
1. **Empirical gains are modest and not state-of-the-art**: While the proposed method shows slight improvements over static low-rank baselines (e.g., Palu, G-LRD), the absolute performance remains well below stronger compression methods such as quantization and key-value eviction. Appendix Tables 6–9 show that 4-bit quantization and eviction-based caches achieve substantially higher accuracy at comparable or greater compression ratios. Yet the paper does not provide a clear justification for preferring low-rank projection methods if the end goal is efficient KV caching. The authors write that their approach is “motivated by the philosophy of reducing information loss through low-rank projection,” but I don't see a reason we should be holding onto low-rank projections if the performance is not state-of-the-art.
Furthermore, I believe the reported KIVI results appear to use incorrect compression ratios (correct me if I am mistaken here). In Table 6, the authors claim compression ratio of KIVI (2-bit) is 2x, but according to the Lexico paper (see their Table 2), KIVI (4-bit) corresponds to ~3× compression and KIVI (2-bit) to 4–5× on LongBench. Also, the baselines for KIVI seems to differ from those in the KIVI paper (see their Table 4). As such, the experimental comparison in the main body is potentially misleading. Overall, the modest improvements over an older low-rank baseline and no improvement over quantization weaken the empirical contribution.
2. **Latency actually increases**: The main text (Fig. 4c) presents a favorable per-layer latency curve, suggesting up to 1.6× speedup over full-KV caching. However, Table 9 in the appendix contradicts this: end-to-end inference latency is longer than both Palu and full-KV caching when measured in a more realistic setup of LongBench decoding. Given that the latency in Figure 4c shows faster speed, the paper is potentially misleading about its latency analysis since the practical runtime is what ultimately matters for model serving. The decrease in throughput (about ~10%) in Table 9 seems a non-negligible amount.

Minor weakness:

3. **Fisher analysis downward trend may just be a methodological artifact**: Section 3 concludes that “high-frequency components are more task-specific and less transferable." However, these “frequency bands” correspond to singular vector indices from an SVD and lower singular-value directions naturally exhibit lower signal-to-noise ratios. Consequently, cosine overlaps will tend to decrease even for identical distributions, producing the same pattern seen in Figure 3. Though one can suggest the possiblity of task-specificity, the observed trend cannot reliably support the stated claim about the necessity of task-specific (or inference-time) updates.
4. **Theory result does not add much**: The theoretical section (Appendix E) attempts to justify low-rank updates by deriving bounds under a stationary data distribution. Yet StreamingKV’s core motivation is test-time adaptation, which is the regime where assuming stationarity is difficult. The derivation does not explain the behavior under distribution drift, so I believe the theoretical claims do not well-support StreamingKV.

**Questions:**

I would like to be upfront and qualify my rating. My evaluation of the paper lies between 2 and 4, and so is 2 at the time of review submission. The questions below are the most important in my re-evaluation.
1. StreamingKV is consistently compared against low-rank compression baselines, but quantization and eviction-based approaches achieve much higher accuracy at similar compression ratios. Can you explain why low-rank projection remains a good design choice for KV compression?
2. Though the method tests on long-input tasks, it is not tested on long-generation tasks like mathematical reasoning. It would be great to see if StreamingKV performs strongly in these, e.g., GSM8K, MATH, or AIME.
3. Can you provide RULER results for quantization-based methods as baselines as well?

Misc:
- I assume full cache in Table 1 means 16 bits?
- L156: "given an input vector $x \in \mathbb{R}^d$", but I assume we should be given a set of $x \sim \mathcal{D}$ here for GHA to learn W to approximate the input distribution.
- L191: pre-trained -> pretrained (for sake of consistency)
- L235: Chang et al. should be \citet{}
- L1121: "Reconsturction" -> "Reconstruction"
- Algorithm 1's L11 is meaningless?

---

### Note · Authors · 2025-11-23

I have read and agree with the venue's withdrawal policy on behalf of myself and my co-authors.